# Expression Analysis of a Novel Oxidoreductase Glutaredoxin 2 in Black Tiger Shrimp, *Penaeus monodon*

**DOI:** 10.3390/antiox11101857

**Published:** 2022-09-20

**Authors:** Rui Fan, Yundong Li, Qibin Yang, Song Jiang, Jianhua Huang, Lishi Yang, Xu Chen, Falin Zhou, Shigui Jiang

**Affiliations:** 1Key Laboratory of South China Sea Fishery Resources Exploitation and Utilization, Ministry of Agriculture and Rural Affairs, South China Sea Fisheries Research Institute, Chinese Academy of Fishery Sciences, Guangzhou 510300, China; 2College of Fisheries and Life Science, Shanghai Ocean University, Shanghai 201306, China; 3Tropical Fishery Research and Development Center, South China Sea Fisheries Research Institute, Chinese Academy of Fishery Sciences, Sanya 572018, China

**Keywords:** glutaredoxin, *Penaeus monodon*, oxidative stress, SNPs

## Abstract

Glutaredoxin (Grx) is a glutathione-dependent oxidoreductase that is an important component of the redox system in organisms. However, there is a serious lack of sequence information and functional validation related to Grx in crustaceans. In this study, a novel Grx was identified in *Penaeus monodon* (*PmGrx2*). The full-length cDNA of *PmGrx2* is 998 bp, with an open reading frame (ORF) of 441 bp, encoding 119 amino acids. Sequence alignment showed that *PmGrx2* had the highest identity with Grx2 of *Penaeus vannamei* at 96.64% and clustered with Grx2 of other crustaceans. Quantitative real-time PCR (qRT-PCR) analysis showed that *PmGrx2* was expressed in all examined tissues, with higher expression levels in the stomach and testis. *PmGrx2* was continuously expressed during development and had the highest expression level in the zygote stage. Both ammonia-N stress and bacterial infection could differentially induce the expression of *PmGrx2* in hepatopancreas and gills. When *PmGrx2* was inhibited, the expression of antioxidant enzymes was suppressed, the degree of apoptosis increased, and the GSH content decreased with the prolongation of ammonia-N stress. Inhibition of *PmGrx2* resulted in shrimp being exposed to a greater risk of oxidative damage. In addition, an SNP locus was screened on the exons of *PmGrx2* that was significantly associated with an ammonia-N-stress-tolerance trait. This study suggests that *PmGrx2* is involved in redox regulation and plays an important role in shrimps’ resistance to marine environmental stresses.

## 1. Introduction

Glutaredoxins (Grxs, also known as thioltransferase), as members of the thioredoxins (Trxs) superfamily, are oxidoreductases that are characterized by their thermal stability and small molecular weight [1,2]. Grxs are indispensable in the redox system of organisms [3]. Grxs were originally discovered in *Escherichia coli*
*lacking* Trxs activity, in 1976 [4]. Typical Grxs fall into two categories according to active site motifs, the dithiol Grxs and the monothiol Grxs, whose active site motifs are Cys-X-X-Cys and Cys-Gly-Phe-Ser, respectively. In the monothiol Grxs, a single Cys residue is present at the n-terminal only. With the assistance of reduced glutathione (GSH), both types of Grxs can reduce disulfide bonds in proteins and convert GSH to oxidized glutathione (GSSH). The difference between the two is that both Cys-active sites of dithiol Grxs are involved in the reaction, whereas monothiol Grxs use only the N-terminal Cys for the reaction. This process is reversible, which implies the important role of Grxs in maintaining intracellular GSH/GSSG ratio homeostasis [5]. Many other types of Grxs are also observed, especially prokaryotes, owing to further research [6]. The Grxs of most types and quantities are currently found in plants, which exhibit a unique active site of Cys-Cys-X-X [7]. In addition, the involvement of Grxs in various physiological and biochemical activities, such as iron–sulfur cluster coordination, resistance to oxidative stress, and cell growth and apoptosis, has been proved [8,9,10]. Instead, there exist numerous studies on prokaryotes, mammals, and plants, while studies on relevant sequences and the functional validation of Grxs in crustaceans are rare.

The black tiger shrimp (*Penaeus monodon*) is an economically promising mariculture crustacean, accounting for about 8% of the crustacean farming industry [11]. However, during the culture process, black tiger shrimps are exposed to oxidative stress damage from biotic or abiotic stressors, leading to high mortality rates from time to time and subsequent significant economic losses [12]. Studies on the expression responses of relevant genes in black tiger shrimps under environmental stress and pathogen infection can enhance shrimp culture management and fill the gap in crustacean-related research.

Therefore, this paper cloned the full-length cDNA of *P. monodon* Grx2 (*PmGrx2*) and elaborated its expression during development and in various tissues. In the hope of clarifying the involvement of *PmGrx2* in other physiological activities in shrimps, the authors selected the most common ammonia-N stress factor and various pathogenic bacteria in the aquatic survival environment and observed the expressions of *PmGrx2* in these conditions and the physiological changes in shrimps after *PmGrx2* was disrupted. Moreover, the SNPs loci of *PmGrx2* related to the ammonia-N-stress-tolerance trait were screened. The study contributes to the future exploration of *PmGrx2’*s role in the regulation of redox homeostasis in shrimp.

## 2. Materials and Methods

### 2.1. Materials

The authors obtained the shrimps from the experimental base of South China Sea Fisheries Research Institute, Shenzhen, Guangdong, China. Shrimps that were 7.01 ± 0.8 g in weight and 5.12 ± 0.69 cm in length underwent RNA interference. Shrimps that were 1.53 ± 0.36 g and 2.11 ± 0.24 cm were used for SNPs analysis. The indicators of the remaining were 15.08 ± 1.20 g and 12.37 ± 0.57 cm. The shrimps were stored in plastic cylinders with aerated filtered seawater for a week (salinity 29‰, temperature 26 ± 2 °C, and pH 7.5–7.8) and were fed once a day with compound feed in the temporary feeding period until 24 h before use.

### 2.2. Materials Collection

The *PmGrx2* expression test required fourteen tissues, while that of *PmGrx2* during development required samples collection under fourteen phases from oosperm to post-larvae [13]. The samples were kept in RNALater solution (Ambion, Austin, TX, USA) at 4 °C for one day and later at −80 °C till being used.

### 2.3. Ammonia-N Stress

As for the testing of *PmGrx2* expression under ammonia-N stress, an acute ammonia-N stress pre-experiment was utilized to study the 96 h median lethal concentration (96 h LC50) and safe concentration (SC) of 180 shrimps. Groups of 30 shrimps were preserved separately in 200 L of filtered seawater, which was aerated constantly through an air stone and refreshed daily. The NH_4_Cl was applied to seawater of per cylinder to prepare ammonia-N in six concentrations, covering 0, 20, 40, 60, 80, and 100 mg/L. The pre-experiment recorded mortality every 3 h, and the 96 h LC50 and SC computed by Linear Regression after completion stood at 29.47 and 2.95 mg/L, respectively [14,15].

The actual experiment and pre-experiment conducted under similar conditions employed shrimps of alike size, which randomly fell into 3 groups (control, 96 h LC50, and SC), each having 3 subgroups consisting of 30 shrimps per. The survival of shrimps was recorded every 3 h without feed supply, and the dead were immediately removed from the container. Three were picked for duplication at different time points, and their hepatopancreas and gills were dissected and preserved in the RNALater solution (Ambion, USA) at 4 °C for one day and then at −80 °C till being used.

### 2.4. In Vivo Experiment

The *PmGrx2* expression test following the immune challenge randomly divided the samples into 4 groups, each having 3 subgroups consisting of 25 shrimps each. The bacteria mentioned were obtained from Key Laboratory of South China Sea Fishery Resources Extraction & Utilization. Samples of the control group were previously injected with 100 μL sterile phosphate buffer solution (PBS, pH 7.4), while those of other groups with 100 μL (1.0 × 10^8^ CFU/mL) of corresponding bacterial solution, respectively [16]. All were intramuscularly injected at the second abdominal segment. Similarly, the survival was logged every 3 h, and the dead were immediately taken out of the container. Three were selected at different time points, and their hepatopancreas and gills were dissected and kept by RNALater solution (Ambion, USA) at 4 °C for one day and later at −80 °C till being used.

### 2.5. Extraction of RNA and Synthesis of cDNA

The HiPure Fibrous RNA Plus Kit (Magen, Guangzhou, China) was used to obtain the total RNA of the samples in quantification experiments, and the RNeasy Mini Kit (Qiagen, Hilden, Germany) was adopted to obtain the total RNA of the remaining samples. Moreover, a NanoDrop 2000 device (Thermo, Waltham, MA, USA) was utilized to explore the ultraviolet absorbance ratio at 260/280 nm, thus determining the purity and quantity of the total RNA, while 1 percent agarose gel electrophoresis was also taken to evaluate its integrity. The RNA was synthesized into the corresponding cDNA, which was stored under −80 °C until it was used.

### 2.6. Cloning the Full-Length cDNA of PmGrx2

*PmGrx2* was screened for partial fragments and identified through the NCBI database BLAST in the cDNA library of *P. monodon* in the laboratory, upon which the primers were established through Premier 6.0. The PCR program, which facilitated the expansion of *PmGrx2* open reading frame (ORF), underwent three procedures: 95 °C (3 min), 35 cycles of 95 °C (15 s), 55 °C (15 s), 72 °C (15 s), and 72 °C (5 min). In addition, the PCR amplification was completed in 25 μL of reaction mixture made up of forward primer (1 μL), reverse primer (1 μL), cDNA template (1 μL), double-distilled water (12.5 μL), and 2 × Taq Plus Master Mix Ⅱ (12.5 μL) (Vazyme, Nanjing, China). Nested PCR primers featuring 3’ and 5’ ends end were generated by Premier 6.0 based on *PmGrx2* ORF. The procedures of the PCR program referred to the literature [11].

All PCR products were cloned into pEASY^®^-T1 Cloning Vector (TransGen, Beijing, China), and a positive monoclonal colony was sequenced. RACE technology was taken to calculate the full-length cDNA of *PmGrx2*, while Ruibiotech (Guangzhou, China) was used for the synthesis of primers and the sequencing of PCR products. Appendix A list the primers above.

### 2.7. mRNA Expression by Quantitative Real-Time PCR (qRT-PCR) Analysis

This study employed qRT-PCR to detect the expression of *PmGrx2* and related genes of samples in the experiment, Premier 6.0. to construct the qRT-PCR primers, and Elongation factor 1α (EF-1α) to represent the reference gene. All primers were synthesized by Ruibiotech (Guangzhou, China) and are demonstrated in Appendix A. A Roche Light Cycler^®^480II contributed to the qRT-PCR, and the specific procedures referred to the literature [11].

### 2.8. Ammonia-N Stress on PmGrx2-Interfered Shrimps

The T7 RiboMAX Express RNAi kit (Promega, Madisonm, WI, USA) was taken to synthesize the double-stranded RNA (dsRNA) for *PmGrx2* (dsGrx2) and green fluorescent protein (GFP, as a non-specific negative control) gene (dsGFP). Appendix A demonstrates the primers adopted for the synthetization. The ultraviolet absorbance ratio at 260/280 nm obtained through a NanoDrop 2000 device (Thermo, USA) promoted the calculation of dsRNA’s purity and quantity, while 1 percent agarose gel electrophoresis shed a light on its integrity. The dsRNA was kept under −80 °C until used.

The *PmGrx2* expression was initially explored by qRT-PCR to clarify the interference efficiency of dsGrx2. The dsRNA was diluted to 1 μg/μL through PBS buffer prior to in vivo injection. The samples were categorized into two groups of three replicates each, with five shrimps in each replicate. The second ventral segment was injected with dsGrx2 or dsGFP dilution (3 μg per gram of shrimp weight), with the former serving as the experimental group and the latter as the control group. Three of each replicate were picked 24 h after injection, and their hepatopancreas were gathered; then the same procedures as in Section 2.7 were followed.

Given dsRNA’s significant interference efficiency, the shrimps were again categorized into two groups with three replicates of 40 shrimps each, and injections were conducted as above. Twenty-four hours after injection, shrimps injected with dsGrx2 or dsGFP experienced acute ammonia-N stress that lasted for 48 h, using 96 h LC50. Three shrimps of each replicate were taken at each time, and the hepatopancreas was gathered. One half was preserved in RNALater solution (Ambion, USA) at 4 °C for one day and later at −80 °C till being used, while the remaining was kept directly in liquid nitrogen till being used.

The steps in Section 2.5 and Section 2.7 were conducted to explore the relative expression of related genes, whose primers are demonstrated in Appendix A. Given the identity of Grx2 as a glutathione-dependent oxidoreductase, shrimps directly preserved by liquid nitrogen were explored through a reduced glutathione (GSH) assay kit (Nanjing Jiancheng Bioengineering Institute, Nanjing, China), and three replicates of each shrimp were detected and statistically probed according to the instruction.

### 2.9. Correlation Analysis between the SNPs of PmGrx2 and Ammonia-N-Stress-Tolerance Trait

The ammonia-N-stress experiment adopted 400 healthy juvenile shrimps and followed the approach mentioned in Section 2.3. The first 40 shrimps that died belonged to the sensitive group, while the last 40 and those still alive were the resistant group. They were kept in ethanol for DNA extraction through MagPure Tissue/Blood DNA LQ Kit (Magen, Guangzhou, China).

The NCBI database was utilized to predict the exonic region of *PmGrx2*, upon which Premier 6.0 designed primers that spanned the exonic regions to amplify the samples’ DNA order. The PCR reaction solution included forward primer (1 µL), reverse primer (1 µL), DNA template (2 µL), and mix green (21 µL) (Tsingke, Beijing, China). The three PCR procedures included 98 °C for 2 min, 35 cycles of 98 °C (10 s), 55 °C (15 s), 72 °C (1 min), and 72 °C (10 min). Tsingke Biotechnology undertook the synthesis of primers and the sequencing of PCR products (Guangzhou, China). The details are given in Appendix A.

DNAMAN and Chromas were adopted to elucidate DNA sequences of each shrimp, thus acknowledging the SNPs of *PmGrx2*. WPS office was used to calculate genetic parameters. An χ2 test was conducted with SPSS 26.0 to explore how the SNPs of *PmGrx2* are correlated to the ammonia-N-stress-tolerance trait, and *p* < 0.05 indicated a significant difference.

## 3. Results

### 3.1. Identification and Characterization of the PmGrx2 Nucleotide Sequence

The cDNA of *PmGrx2* (Appendix A; GenBank No. ON368189) is 998 bp in full length, covering 179 bp 5’-UTR, 459 bp 3’-UTR, a poly-A tail, and 360 bp ORF, as well as encoding a 119 amino acid putative protein. The putative *PmGrx2* protein, which was 12.87 kDa and covered only one Grx structural domain, had an isoelectric point of 8.51 theoretically. The active site motif of the *PmGrx2* domain was as typical as other known Grx2 proteins, C-P-Y-C (39–42 aa). All of these supported the inclusion of *PmGrx2* into the dithiol Grxs family. In the putative secondary structure, the α-helix, extended strand, β-turn, and random coil accounted for 42.02%, 15.97%, 8.40%, and 33.61%, respectively. Appendix A illustrates the tertiary structure.

The paper compared the amino acid sequences of Grx2 of seven crustacean species, most of which were conserved, and all had active site C-P-Y-C (Appendix A). *PmGrx2* had the highest identity with Grx2 from *Penaeus vannamei* (96.64%). The homology parameters of Grx2 and PmGrx2 for the species involved in the comparison are shown in Appendix A. The amino acid sequences of fifty-four Grxs were chosen, and the minimal evolution of MEGA-X was employed to create a phylogenetic tree. Grx3 and Grx5 were clustered together, Grx2 from plants and bacteria was clustered separately, and Grx2 from the remaining species was clustered together (Appendix A). Grx from crustaceans clustered closely, with *PmGrx2* being most closely related to Grx2 from *P**. vannamei* and *Penaeus japonicus*. The scientific names and accession numbers of all the above species are shown in Appendix A.

Given the shortage of relevant sequence information and validation concerning crustaceans, the amino acid sequence of human Grx2 was selected for protein interaction analysis. The predicted results are shown in Figure 1. Among the proteins that have major interactions with Grx2, Grx3 and Grx5 belong to the same Grxs family as Grx2. Glutathione reductase (GSR) and glutathione peroxidase (GPX) are glutathione-dependent enzymes, and thioredoxin reductase 2 (TXNRD2) and thioredoxin (TXN) are key enzymes of the thioredoxin system. These proteins are important components of the eukaryotic antioxidant system, along with catalase (CAT). In addition, RNA-binding motif protein 28 (RBM28), isochorismatase 1 (ISOC1), and neural precursor-cell-expressed developmentally downregulated 8 (NEDD8) also interact closely with Grx2. The names and websites of the analysis software above are shown in Appendix A.

### 3.2. mRNA Expression of PmGrx2 in Different Tissues and Developmental Stage

*PmGrx2* was expressed in all 14 tissues of the shrimp and was significantly more abundant in the stomach and testis than in other tissues (*p* < 0.05). *PmGrx2* was less abundant in the ovary, brain, eyestalk nerves, and abdominal nerves (Figure 2).

The expression of *PmGrx2* was constantly observed along the evolution of shrimp from zygote to post-larva, which obviously outnumbered in the zygote period compared with other periods (*p* < 0.05). As shrimp development progressed to the nauplius stage, the *PmGrx2* expression increased as a whole. In the zoea and mysis stages, the expression of *PmGrx2* decreased and fluctuated, and it only temporarily surged in zoea III (Figure 2B).

### 3.3. Exploration of PmGrx2 Transcription of Hepatopancreas and Gills on the Heels of Bacterial Challenge

Figure 3A validated the much higher expression of *PmGrx2* in the hepatopancreas compared with the control (*p* < 0.05) after the injection of *Staphylococcus aureus,* as well as its peak at 72 h after injection. Such expression, however, failed to jump after injection of two Gram-negative bacteria separately, especially *Vibrio anguillarum*. After injection of *Vibrio Harveyi*, it only picked up a certain amount at 24 h post-injection, followed by a decline. Figure 3B denies the exact similarity of PmGrx2’s expression profiles between the gill and the hepatopancreas after bacterial injection. However, *S. aureus* still significantly elicited a response of *PmGrx2* in the gill (*p* < 0.05), just not as violently as *PmGrx2* in the hepatopancreas. After separate injections of the two Gram-negative bacteria, no jump of *PmGrx2* expression was observed in the gill compared to the control.

### 3.4. Exploration of PmGrx2 Transcription of Hepatopancreas and Gill under Ammonia-N Stress

Figure 4A confirms the similarity of the *PmGrx2’*s expression pattern in the two experimental groups under ammonia-N stress, and both peaked at 12 h (*p* < 0.05). No jump in the *PmGrx**2’*s expression was observed in both experimental groups with increasing time compared with the control (*p* < 0.05), but its expression profile of the gill at 96 h following ammonia-N stress differed greatly from that in the hepatopancreas (Figure 4B). *PmGrx2’*s expression in the gill of the 96 h SC group outnumbered the control group within 12 h following the stress (*p* < 0.05), peaked at 6 h, and was not on par with the control group as the duration of ammonia-N stress prolonged. In contrast, its expression of the 96 h LC50 group never jumped (*p* < 0.05). *PmGrx2’*s expression was much lower compared with the control (*p* < 0.05) 12 h following stress.

### 3.5. The Interference Efficiency of dsGrx2 in the Hepatopancreas

Figure 5A proves dsGrx2’s role in sharply decreasing *PmGrx2’*s expression in the hepatopancreas (*p* < 0.05). Figure 5B reveals the application of ammonia-N stress into both groups 24 h following dsRNA injection. As to *PmGrx2’*s expression within 48 h following stress (*p* < 0.05), the dsGFP-injected group sharply outnumbered dsGrx2-injected group, indicating the powerful inhibitory effect of dsGrx2 within 72 h following injection.

### 3.6. GSH Content and mRNA Expression of Related Genes in the Hepatopancreas of dsRNA-Injected Shrimps under Ammonia-N Stress

Figure 6A demonstrates the gradual climb of *PmTrx*’s expression in the dsGFP-injected group, its peak at 12 h following ammonia-N stress, and the following gradual decrease. Such expression of the group injected with dsGrx2 also increased gradually and then decreased after a period of fluctuation. The expression of *PmTrx* in the dsGrx2-injected group outnumbered the dsGFP-injected counterpart at 6 h and 24 h following stress (*p* < 0.05).

In regard to *PmPrx1’*s expression, Figure 6B reveals its increase at 3 h under stress, and the following decrease and fluctuation at a similar level in the group injected with dsGFP. The one in the dsGrx2-injected group with the same stress, however, dramatically fluctuated and peaked at 6 h. The expression of *PmPrx1* of the two groups exhibited an opposite trend and differed significantly after 3 h, 6 h, and 24 h under the stress (*p* < 0.05).

Figure 6C validates both groups’ similar expression tendency of *PmCAT* within 12 h after the stress, which first increased and then decreased. However, starting from 24 h, the dsGrx2-injected group presented a much lower expression (*p* < 0.05).

Figure 6D shows the similar expression tendency of *PmCYC* in both groups within 12 h following the stress, as well as the much higher expression in the dsGrx2-injected group (*p* < 0.05). One day later, the expression of the dsGrx2-injected group declined, and it decreased abruptly at 48 h (*p* < 0.01). In contrast, the expression of the dsGFP-injected counterpart gradually increased.

Figure 6E illustrates *PmIAP*’s irregularly fluctuated expression of the dsGFP-injected group within 48 h following the stress. The dsGrx2-injected group showed an increase and then a decrease in the expression of *PmIAP*, and then it reached a maximum at 6 h (*p* < 0.05).

Figure 6F illustrates the groups’ similar trend of GSH content in 48 h following the stress in the hepatopancreas. The GSH content was the lowest in both groups at 48 h, while that of the dsGrx2-injected group was much lower compared to the group injected with dsGFP (*p* < 0.05).

### 3.7. Correlation Analysis between the SNPs of PmGrx2 and Ammonia-N-Stress-Tolerance Trait

This paper found two SNPs on the exons of *PmGrx2*, whose information and sequencing maps are demonstrated in Table 1 and Figure 7, respectively. According to Table 2, which reveals polymorphic parameters, *PmGrx2*-E3371 had a Ho of 0.0125, He of 0.0125, Ne of 1.0126, and MAF of 0.0063; the corresponding values for *PmGrx2*-E3398 were 0.0000, 0.0722, 1.0778, and 0.0375, respectively. Both *PmGrx2*-E3371 and *PmGrx2*-E3398 exhibited low polymorphism (PIC < 0.2500). The HWE results showed that *PmGrx2*-E3398 deviated significantly from HWE, while *PmGrx2*-E3371 did not. Table 3 displays the relation between the SNPs and ammonia-N-stress-tolerance trait. *PmGrx2*-E3398 was significantly correlated with the latter (*p* < 0.05), while *PmGrx2*-E3371 was not.

## 4. Discussion

For the first time, the authors succeed in cloning the full-length cDNA of a new glutaredoxin, *PmGrx2*, in crustaceans. The only active site motif that is present among the predicted motifs was the Grx structural domain of C-P-Y-C, a typical dithiol Grx structural domain, which supported the inclusion of *PmGrx2* into the Grxs family [5]. This paper picked the amino acid sequences of Grx2 of seven crustacean species for multiple sequence alignment, and all of them had C-P-Y-C in their sequences. Phylogenetic analysis showed that the monothiol Grx3 and Grx5 clustered together, while the dithiol Grx2 clustered separately. Among them, *PmGrx2* converged to the one of crustacean Grx2. The above indicates that the Grx active sites are highly conserved among different species. However, Grxs are a large family with distinct segregation characteristics, both from a species-classification point of view and from a Grxs-classification point of view. For example, plants are known to possess dozens of Grxs, far more than other eukaryotes [18]. In recent years, new types of Grxs have been found in prokaryotes [6]. The abovementioned suggests that different types of Grxs may have distinct functions, and the taxonomic diversity they exhibit guides us to investigate and add new elements in different species.

Of the proteins predicted to interact with Grx2, most are core members of the eukaryotic redox system. This suggests that the fundamental role of Grx2 is to maintain redox homeostasis in the organism. In addition, RBM28 is an intracellular host factor that can interact directly with RNA viruses [19]. ISOC1 is involved in intracellular pathogen recognition and clearance [20]. NEDD8 is vital for the organism’s defense mechanism against proteotoxicity [21]. The three proteins matter a lot in the organism’s reaction to pathogen infection and immune activation. The predicted result that Grx2 interacts with them indicates the potential involvement of *PmGrx2* in the innate immune process in shrimp.

*PmGrx2* was significantly more expressed in the stomach and testis than in other tissues. On the one hand, the stomach, as an important digestive organ of shrimp, undertakes functions such as grinding food, a process that consumes energy and generates a variety of oxygen metabolites [22]. On the other hand, the gastric epithelium acts as an external barrier to the digestive system, and it is crucial to maintain its redox homeostasis [23]. These may be the reasons for the elevated expression of *PmGrx**2* in the stomach. Previous studies have shown that Grx2 expression levels are elevated during human sperm maturation. Moreover, Grx2b and Grx2c are testis-specific protein isoforms. It is hypothesized that Grx2 may be involved in the generation of disulfide bonds during sperm maturation [24]. In addition, a redox homeostatic system exists in reproductive tissues of animals, and the antioxidant mechanism is activated when stimulated by ROS, thus reducing the extent of oxidative damage to spermatozoa [25]. Therefore, *PmGrx2* may play the same role in the stomach and testis of shrimp.

*PmGrx2* was expressed throughout the early growth of black tiger shrimps, which is complex and rapid, and employs mitochondria as feed [26]. In zebrafish, Grx2 was shown to participate in the coordination of intracellular iron–sulfur clusters [8]. Considering the vital role of the iron–sulfur cluster in the mitochondrial respiratory chain, *ZfGrx2* matters much as to the energy supply of mitochondria and is involved in regulating vascular, heart, and brain growth in zebrafish embryos [27,28,29,30]. All of these indicate *PmGrx2’*s similar function in driving their early development.

*PmGrx2* in both hepatopancreas and gills responded strongly to *S. aureus* (Gram-positive bacteria) after injection with different species of bacteria. In contrast, both Gram-negative bacteria had almost no effect on the expression of *PmGrx2* compared to the control group. This indicates *PmGrx2’*s part in innate immune process of shrimp against pathogenic bacteria, but with different sensitivities to various species of pathogens. Since Gram-positive bacteria tend to produce exotoxins and do not have an outer membrane formed by lipopolysaccharides. Gram-negative bacteria tend to produce endotoxins and have capsules and mucus layers that cover the outer membrane, a characteristic that makes it easier for Gram-negative bacteria to invade organisms latently. In fact, it is often Gram-negative bacteria that cause mortality in aquatic animal populations during aquaculture. Gram-negative bacteria do not elicit *PmGrx2’*s response in the gills, and this is one of the favorable conditions for their invasion of the organism. Meanwhile, the hepatopancreas was considered to be an immune organ of the highest significance in shrimp [31]. *PmGrx2’*s expression in the gills of the experimental group failed to be as strong as that in the hepatopancreas in comparison with the control, despite the higher expression of *PmGrx2*.

Ammonia-N refers to an ordinary environmental stressor in aquaculture processes. Excess ammonia-N induced excessive production of ROS/RNS, which caused oxidative damage and inflammatory effects in aquatic organisms [32,33,34]. *PmGrx2’*s expression in the 96 h SC group increased to different degrees following ammonia-N stress in comparison with the control group, and this was true for both hepatopancreas and gills. However, such expression in the 96 h LC50 group was only increased in the hepatopancreas, and in the gills, it was always lower compared with the control group. It is hypothesized that, since the gills are the first line of defense in contact with the aquatic environment, high concentrations of ammonia-N potentially undermine the gills in shrimp [15,35]. In *P. vannamei*, Grx2’s expression of hepatopancreas and gills increased similarly following ammonia-N stress [36]. All of these suggest that *PmGrx2* participates in the resistance of shrimp to stress.

In regard to the part of *PmGrx2* in the resistance of shrimp to ammonia-N stress, dsRNA was employed to interfere *PmGrx2’*s expression, which supports its powerful interference role at least 72 h following injection. *PmTrx* and *PmPrx1*, also included in the Trx superfamily as *PmGrx2*, were picked for expression elaboration following the suppression of *PmGrx2’*s expression. In the hepatopancreas, *PmTrx*’s expression of the dsGrx2-injected group started to outnumber the dsGFP-injected group at 6 h following stress and was significant at 6 h and 24 h. *PmPrx1’*s expression in the dsGrx2-injected group, however, performed the opposite trend compared to the dsGFP-injected group under ammonia-N stress. Previously, it was shown that Grx could functionally supplement *E. coli* Trx mutant strains [4]. It is hypothesized that functionally similar enzymes in the same family complement each other, and the functional decline of one side surely causes the expression upregulation of the other side to compensate for the functional deficiency of the organism. This is the reason why Grx was originally called the Trx backup system [9]. With longer stress time, the dsGrx2-injected group presented much lower *PmCAT*’s expression compared with the dsGFP-injected group. *PmCAT* is a peroxidase, and the hike of its expression always oxidatively damages the organism [37]. As for *PmCYC*, an enzyme that was vital in apoptosis, its expression remained high in the dsGrx2-injected group compared to the dsGFP-injected group from 6 h to 24 h after stress [38]. IAP is an anti-apoptotic enzyme [39]. At the beginning of ammonia-N stress, *PmIAP*’s expression in the group with dsGrx2 gradually increased; however, from 12 h after stress, the expression was lower compared with the dsGFP-injected group. Meanwhile, the GSH content in hepatopancreas dropped. The abovementioned indicated the organism’s upregulation of genes with alike functions to fight against ammonia-N stress in the presence of inhibited *PmGrx2*, some antioxidant enzymes’ lower expression, and a higher degree of apoptosis. When *PmGrx2* was inhibited, the resistance of shrimp was weakened in the face of oxidative damage.

This study firstly adopted direct sequencing in *PmGrx2* for locus typing and obtained two SNPs loci, among which *PmGrx2*-E3398 significantly deviated from the Hardy–Weinberg equilibrium (*p* < 0.05). Allele frequency is swayed by manual selection, as well as extreme samples [11]. Both *PmGrx2*-E3371 and *PmGrx2*-E3398 are located on the ORF. Among them, *PmGrx2*-E3398 is a synonymous mutation, while *PmGrx2*-E3371 is a missense mutation, which would result in the mutation of glutamine (Gln) to histidine (His). Both Gln and His are polar amino acids. In the present study, *PmGrx2*-E3398 was highly related to the ammonia-N-stress-tolerance trait (*p* < 0.05). Since this mutation did not involve alteration of the encoded protein, the specific mode of its effect requires further investigation. Despite the long-standing neutral function of synonymous mutations, studies in recent years have demonstrated that synonymous mutations affect the efficiency and stability of transcription and translation of genes [40,41,42]. *PmGrx2*-E3398 could be developed as a molecular marker for breeding shrimp tolerant to stress.

## 5. Conclusions

The cDNA of *PmGrx2* was cloned and included into Grxs family. *PmGrx2* was proved to express most in the stomach and testis of shrimp, and such an expression can be found in the early developmental stages. The expression of *PmGrx2* in the hepatopancreas and gills was induced owing to ammonia-N stress and bacterial infection, indicating the essential part of *PmGrx2* in the defense mechanism against environmental stress and pathogen infection. The inhibition of *PmGrx2* expression under the ammonia-N stress resulted in a greater risk of oxidative damage to shrimp. Moreover, this paper acknowledged SNPs in the exon region of *PmGrx2*, explored their relation to the ammonia-N-stress-tolerance trait, and successfully screened an SNP locus whose relationship with the ammonia-N-stress-tolerance trait was determined to be significant. This paper lays the foundation for studies on the involvement of Grx2 in crustacean resistance to oxidative stress.

## Figures and Tables

**Figure 1 antioxidants-11-01857-f001:**
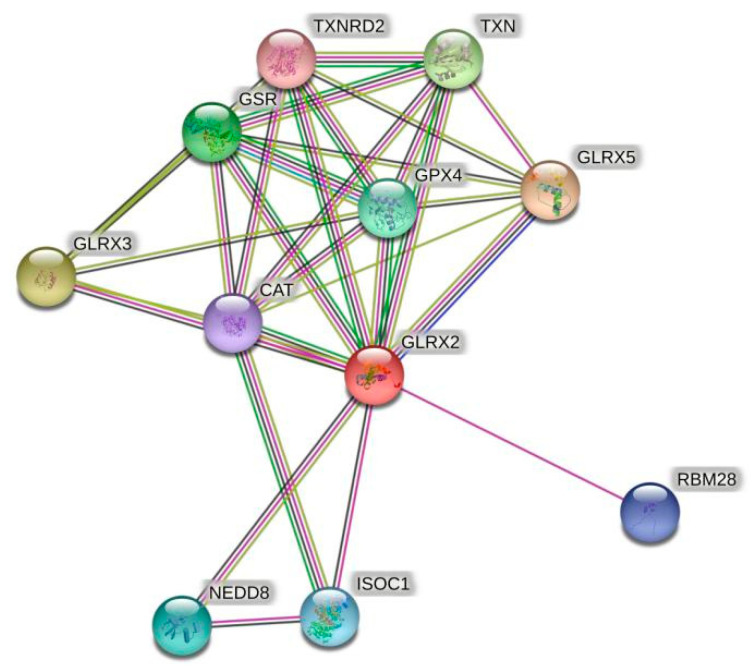
Grx2 protein interactions networks. The red dots represent the target protein Grx2. Other colored dots represent proteins that interact with Grx2. The names of all proteins are abbreviated to the top right of the dots.

**Figure 2 antioxidants-11-01857-f002:**
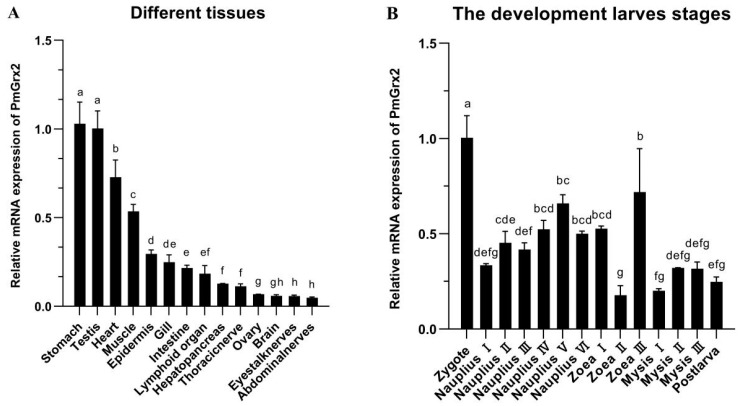
(**A**) mRNA expression levels of *PmGrx2* in different tissues. The data are presented as mean ± SD (n = 3). Different letters show significant differences (*p* < 0.05). (**B**) mRNA expression levels of PmGrx2 during the developmental period. The data are presented as mean ± SD (n = 3). Different letters are used to show significant differences (*p* < 0.05).

**Figure 3 antioxidants-11-01857-f003:**
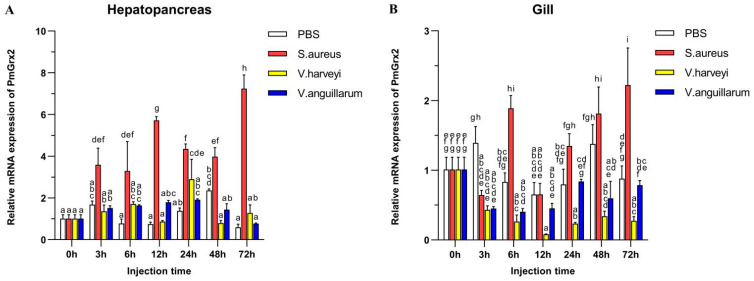
mRNA expression levels of *PmGrx2* in the hepatopancreas (**A**) and gill (**B**) at different time intervals after multiple pathogenic bacteria injections. The data are presented as mean ± SD (n = 3). Different letters are used to show significant differences (*p* < 0.05).

**Figure 4 antioxidants-11-01857-f004:**
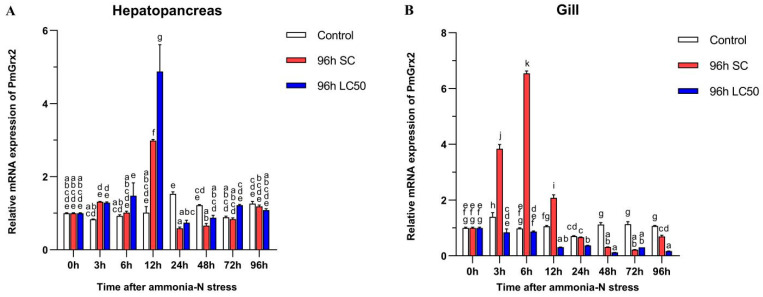
mRNA expression levels of *PmGrx2* in the hepatopancreas (**A**) and gill (**B**) at different time intervals, under different concentrations of ammonia-N stress. The data are presented as mean ± SD (n = 3). Different letters are used to show significant differences (*p* < 0.05).

**Figure 5 antioxidants-11-01857-f005:**
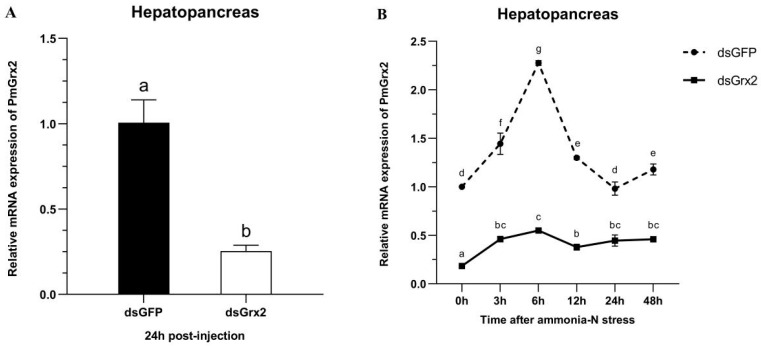
(**A**) mRNA expression levels of *PmGrx2* in the hepatopancreas at 24 h after dsRNA injection. (**B**) Expression profile of *PmGrx2* in the hepatopancreas of dsRNA-injected shrimps within 48 h under ammonia-N stress. The data are presented as mean ± SD (n = 3). Different letters show significant differences (*p* < 0.05).

**Figure 6 antioxidants-11-01857-f006:**
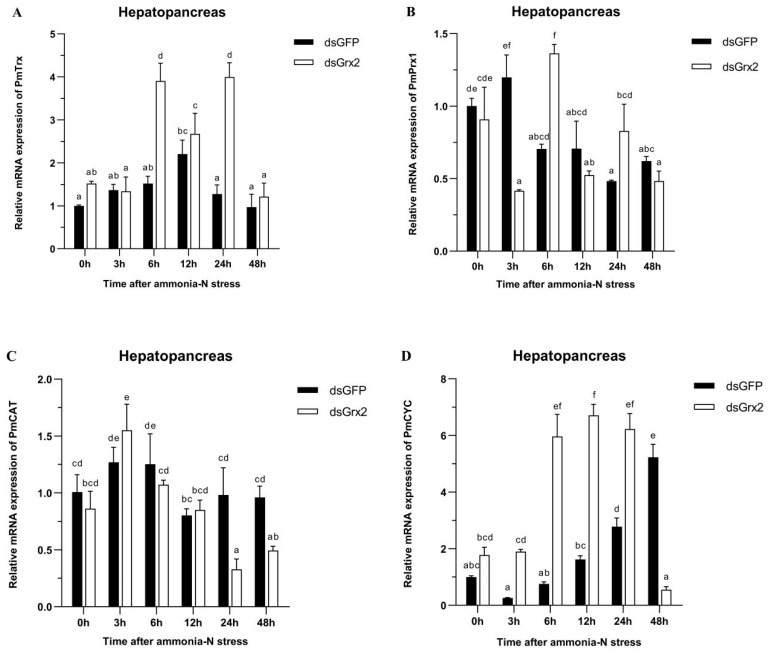
Expression profiles of five genes (*PmTrx* (**A**), *PmPrx1* (**B**), *PmCAT* (**C**), *PmCYC* (**D**), and *PmIAP* (**E**)) and changes in GSH content (**F**) in the hepatopancreas of dsRNA-injected shrimps under ammonia-N stress. The data are presented as mean ± SD (n = 3). Different letters show significant differences (*p* < 0.05).

**Figure 7 antioxidants-11-01857-f007:**
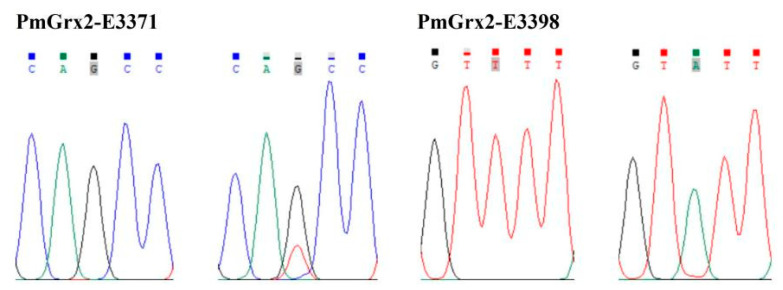
Sequencing maps of the SNPs of *PmGrx2*.

**Table 1 antioxidants-11-01857-t001:** Specific information of the SNPs of *PmGrx2*.

SNPs	Position	Type of Base Mutation	Type of Protein Mutation
PmGrx2-E3371	Exon 3 (371 bp)	c.192G > T	Mm p.Gln64His
PmGrx2-E3398	Exon 3 (398 bp)	c.219T > A	Sm p.Val73=

Mm, missense mutation; Sm, synonymous mutation; Meaning of the symbols in Table 2 [17].

**Table 2 antioxidants-11-01857-t002:** Polymorphic parameters of the SNPs of *PmGrx2*.

SNPs	H_o_	H_e_	N_e_	MAF	PIC	HWE
PmGrx2-E3371	0.0125	0.0124	1.0126	0.0063	0.0123	0.9984
PmGrx2-E3398	0.0000	0.0722	1.0778	0.0375	0.0696	0.0000

SNPs, single nucleotide polymorphisms; H_o_, observed heterozygosity; H_e_, expected heterozygosity; N_e_, effective number; MAF, minimum allele frequency; PIC, polymorphism information content; HWE, Hardy–Weinberg equilibrium.

**Table 3 antioxidants-11-01857-t003:** Correlation analysis between the SNPs of *PmGrx2* and ammonia-N-stress-tolerance trait.

SNPs	Genotype	Genotype Frequencies		χ2 Value	*p*-Value
		Sensitivity	Resistance		
PmGrx2-E3371	GG	1.0000	0.9750	2.0006	0.1572
	TG	0.0000	0.0250		
PmGrx2-E3398	AA	0.0750	0.0000	6.0163	0.0142 *
	TT	0.9250	1.0000		

* Denotes significant correlation (*p* < 0.05).

## Data Availability

The data presented in this study are available in the article and Appendix A.

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
