# Peer review of "Expression Analysis of a Novel Oxidoreductase Glutaredoxin 2 in Black Tiger Shrimp, *Penaeus monodon"

_antioxidants, 2022, doi:10.3390/antiox11101857_

Round 1

Reviewer 1 Report

The authors report on a novel glutaredoxin identified in PmGrs", i.e. Penaeus monodon and establish its high homology with other crustacean glutaredoxins.

The new Grx is expressed particularly in stomach and testis. Its expression in hepatopancreas and gills was induced by ammonia N-stress and bacterial infection. Importantly, its inhibition suppresses the expression of antioxidant enzymes increasing the risk of exposure of the shrimp to oxidative stress. 

The paper is clearly written, and the topic is relevant and suitable for publication in Antioxidants. This Reviewer recommends expanding a bit more the Introduction, indulging on the mechanism of Grx in the antioxidant endogenous defence. The resolution of all figures is poor and must be optimized. In addition, the use of colors is encouraged: most of the readers nowadays avoid printing and colors make the interpretation more straightforward. I have no particular concern on this work and thus suggest publication after the implementation of the requested changes.

Reviewer 2 Report

Dear Authors, the article is interesting with high impact of novelty. The structure of manuscript is correct however some elements could be improved.

 Comments;

 1.       In the title authors indicate “Molecular characterization…” however this part of analysis is very short in main body of manuscript and was mainly moved to supplementary section. The section 3.1 should be presented with the GRX model that very well fit to described results. Moreover conservation in enzyme structure can be presented with using a very nice tool like ConSurf https://consurf.tau.ac.il/overview.php. The name of subsection should be changed because the bioinformatic has a very wide meaning and also next analysis like expression profiles can be analyzed or were analyzed with use bioinformatic tools.

2.       Line 38-42; The description of GRX classification should be corrected. The classification is done according to occurrence of specific amino acid motifs in active site region authors should use this kind of words. Now it seem to that eg. CYS-X-X-CYS is an active site.

3.       In introduction authors could shortly describe mechanism of enzyme action and its specific role in redox homeostasis maintaining not only that it is engaged in this process but what it is responsible for (substrates, coenzymes etc.).

4.       Line 390. The sentence should be corrected now is incomprehensible.

5.       Line 400-412; Why the expression of GRX is higher in stomach and testicles. From actual description it is not clear. Maybe the food digestion and formation of ROS/RNS or other danger derivates requires higher activity of GRX ? In case of testicles high energy consumption during spermatogenesis can be also reason of free radical formation and induce higher response of antioxidant defense system. Please try to discuss it on this way not only compare how it is in other organism.

6.       Why Gram negative bacteria did not influence on GRX expression in gills ? Please try to explain it.

7.       The manuscript need correction by English native speaker.

Reviewer 3 Report

This study is very interesting and well prepared. However, I have major comments as follows:

- There are some typo errors.

- letters of significance are too small.

- There is a high similarity between this work and https://doi.org/10.3389/fmars.2022.909827.

- Table S1 needs to be rewritten for more clarification.

- The bioinformatics analysis of Grx is not precise.

Round 2

Reviewer 2 Report

Dear authors, the manuscript was improved and can be published in Antioxidant.

Reviewer 3 Report

The authors followed the comments.